# Geologically Meaningful $^{40}$Ar/$^{39}$Ar Ages of Altered Biotite from a Polyphase Deformed Shear Zone Obtained by *in Vacuo* Step-Heating Method: A Case Study of the Waziyü Detachment Fault, Northeast China

**Wenbei Shi** [1,2,*], **Fei Wang** [1,2,3,4], **Lin Wu** [1,2], **Liekun Yang** [1,2], **Yinzhi Wang** [1,2] and **Guanghai Shi** [5]

1   State Key Laboratory of Lithospheric Evolution, Institute of Geology and Geophysics, Chinese Academy of Sciences, Beijing 100029, China; wangfei@mail.iggcas.ac.cn (F.W.); wulin08@mail.iggcas.ac.cn (L.W.); liekunyang@mail.iggcas.ac.cn (L.Y.); wangyinzhi@mail.iggcas.ac.cn (Y.W.)

2   Innovation Academy for Earth Science, Chinese Academy of Sciences, Beijing 100029, China

3   CAS Center for Excellence in Tibetan Plateau Earth Sciences, Beijing 100101, China

4   College of Earth and Planetary Sciences, University of Chinese Academy of Sciences, 19(A) Yuquan Rd, Shijingshan District, Beijing 100049, China

5   State Key Laboratory of Geological Processes and Mineral Resources, China University of Geosciences, Beijing 100083, China; shigh@cugb.edu.cn

*   Correspondence: shiwenbei@mail.iggcas.ac.cn

**Abstract:** Discordant biotite $^{40}$Ar/$^{39}$Ar age spectra are commonly reported in the literature. These can be caused by a number of processes related to in vacuo heating, homogenization of the argon distribution, and production of misleadingly flat age spectra. Problematic samples are typically derived from metamorphic belts; thermal overprinting and chloritization are two of the main known causes of disturbed age spectra. Biotite and muscovite of the Waziyü detachment fault, Yiwulüshan metamorphic core complex, Jinzhou, China, yield highly variable $^{40}$Ar/$^{39}$Ar data that hinder reconstruction of their deformation history. We combined mineralogical studies with detailed $^{40}$Ar/$^{39}$Ar dating of biotite, phengitic white mica, and K-feldspar augen from this fault. We infer that argon within the biotite was modified by hydrothermal fluids during fault activity and associated epidotization, chloritization, and muscovitization such that bulk sample step-heating, single grain total fusion, and in situ laser ablation of biotite produced mixed $^{40}$Ar/$^{39}$Ar ages. However, detailed step-heating of biotite shows that this mineral records the ages of cooling and later alteration based on data from a coexisting rigid feldspar porphyroblast and neo-crystallized phengite that record two periods of fault activity at ~120–113 and 18–12 Ma. Our data reveal that the discordant biotite $^{40}$Ar/$^{39}$Ar age spectra might represent a mixed age and that only detailed step-heating methods can extract meaningful geological details of the deformation history of a fault. Therefore, the mineral and the method must be carefully considered if metamorphic or deformed samples are dated.

**Keywords:** Waziyü detachment fault; epidotization; biotite; $^{40}$Ar/$^{39}$Ar dating

## 1. Introduction

Biotite is frequently used in $^{40}$Ar/$^{39}$Ar dating. However, disturbed $^{40}$Ar/$^{39}$Ar biotite age spectra are common, and thus the interpretation of biotite ages to produce a thermochronological framework is challenging. Biotite is a hydrated mineral, and dehydroxylation during in vacuo heating tends to

homogenize radiogenic argon ($^{40}$Ar*) gradients within biotite grains to produce flat age spectra [1]. Disturbed age spectra, such as saddle-, staircase-, and hump-shaped age spectra, which are commonly reported in the literature, indicate that homogenization is not always effective [2–6]. Discordance at temperatures above 600 °C is amplified by dramatic changes in the crystal structure of biotite in response to dehydroxylation and oxidation [7]. Therefore, explanations of the disturbed $^{40}$Ar/$^{39}$Ar age spectra and derivation of meaningful $^{40}$Ar/$^{39}$Ar ages from biotite is a long-standing challenge [2,4,7–12].

York and Loped-Martinez [4] summarized four types of biotite $^{40}$Ar/$^{39}$Ar age spectra as follows: (1) flat plateau age spectra; (2) low-temperature staircase spectra leading to a plateau; (3) staircase spectra at low temperature leading to a dip at medium temperatures and a higher plateau; (4) a mostly flat plateau with a hump within the medium-temperature range. They argued that the third type of age spectra are produced when argon degasses from more than one phase in addition to biotite, such that $^{40}$Ar* related to a later thermal overprint is released separately. However, Lo and Onstott [7] proposed that this type of age spectra is characteristic of chloritized biotite and records recoil of $^{39}$Ar$_K$ ($^{39}$Ar produced from $^{39}$K during the irradiation) from biotite to chlorite layers during neutron irradiation. In this case, argon release during in vacuo step-heating is dominated by chlorite and biotite alternately and produces apparent ages that are younger or older than the cooling age. They concluded that most discordant biotite $^{40}$Ar/$^{39}$Ar spectra can be attributed to this mechanism based on model calculations. The chloritized biotite of Di Vincenzo et al. [10] show very young ages for the low-temperature steps, a hump in the age spectra at medium temperatures, followed by a decrease to slightly younger ages, although these ages are still older than the emplacement age. They attributed these changes in the apparent ages to argon hosted by defects, chlorite, and biotite. However, the total fusion ages are approximately the same as the emplacement age. The real emplacement or cooling age of these samples can only be obtained from pristine domains or by in situ laser ablation techniques. Roberts et al. [9] reached the same conclusion after a comparison of the ages of three variably altered biotite samples by infrared (IR) and ultraviolet (UV) laser techniques. Age gradients within experimentally deformed or thermally overprinted biotite can only be revealed by in situ UV laser dating [13,14].

However, some aspects of these previous studies require further consideration. New research shows that mechanisms other than redistribution of $^{39}$Ar$_K$ by recoil within interlayered chlorite and biotite can explain disturbed age spectra [11]. During metamorphic events, re-equilibrated biotite, and biotite relicts within mica can retain different $^{40}$Ar/$^{39}$Ar ages [15], and it has been shown that mixtures of multiple biotite populations can produce peak–valley–peak-type age spectra [11]. Therefore, disturbed age spectra are produced if biotite grains from different population sources are not separated effectively during sample preparation. Furthermore, if alteration affects grains on length scales less than the spatial resolution of in situ laser ablation techniques, then coexisting biotite populations or argon loss related to alteration will produce mixed spot ages [16]. In addition, biotite alteration under mesothermal conditions produces a range of secondary minerals, such as biotite, muscovite, epidote, and clay minerals [17]. Few studies have investigated the effects of this alteration on $^{40}$Ar/$^{39}$Ar biotite dating.

To understand the effects of alteration on biotite $^{40}$Ar/$^{39}$Ar dating results, it is necessary to know the original cooling age and timing of biotite alteration. The cooling age can be constrained using pristine flakes of biotite [9] or alternative geochronological techniques [10], but the timing of alteration must also be known to optimize the accuracy of the dating procedure. In this paper, we describe a detailed methodological study of $^{40}$Ar/$^{39}$Ar dating of biotite from epidotized and muscovitized samples using traditional and detailed step-heating, laser probe, and total fusion techniques. The results are combined with $^{40}$Ar/$^{39}$Ar dating of K-feldspar and phengitic mica from the same sample to derive insights into biotite $^{40}$Ar/$^{39}$Ar dating systematics.

## 2. Geological Setting

The Yiwulüshan metamorphic core complex (MCC) [18], also known as the Waziyü MCC [19], is located within the eastern Yinshan–Yanshan Orogen. It is bounded to the east, the west, and the

south by the Xialiaohe, the Fuxin–Yixian, and the Bohaiwan basins, respectively (Figure 1A). This area records the Yanshan tectonic transition from compression to extension during the late Mesozoic [20].

The footwall of the Yiwulüshan MCC comprises Archean gneiss, Proterozoic metasedimentary rocks, and plutons emplaced during the Jurassic and the Cretaceous ages. The hanging wall is a syn-extensional rift basin filled by Lower Cretaceous clastic sediments [19]. The master detachment fault is marked by a mylonitic belt on the west of pluton with a foliation that dips at 25°–35° towards 280°–340° and a stretching lineation plunging towards 270°–295° at 5°–24° [21]. Dynamically recrystallized quartz aggregates and brittlely deformed feldspar record deformation at greenschist-facies conditions (>300–350 °C), consistent with temperatures of 300–400 °C derived from the lattice-preferred orientation of quartz [20], and the brittle deformation of feldspar, which indicates temperatures <500 °C [19].

Previous geochronological results indicate that the Lüshan pluton, which is the main pluton within the Yiwulüshan MCC, was emplaced simultaneously with the Guanyindong and the Jianshilazi plutons at ~163–152 Ma, based on zircon U–Pb dating [19,20,22,23]. The Shishan pluton, which is located at the southern end of the Lüshan pluton, yields a younger zircon U–Pb age of 123 Ma; this pluton shows no or weak foliation [22]. A number of biotite samples from the shear zone and the mylonitic belt associated with the Lüshan pluton have been dated by $^{40}Ar/^{39}Ar$ methods. The results fall within a broad age range of 150–97 Ma and commonly show discordant age spectra. The zircon and the apatite fission-track dating results of Ma et al. [18] indicate that the Yiwulüshan MCC experienced a long, slow, cooling history from 81 to 14 Ma, and that exhumation of the Yiwulüshan MCC continued until 5–4 Ma.

Structural analysis indicates that the area experienced two phases of tectonic activity, including a Late Jurassic–Early Cretaceous (~141 Ma) compressional event and an Early Cretaceous (~126 Ma) NW–SE ductile extensional shearing event [20]. Biotite and muscovite $^{40}Ar/^{39}Ar$ ages younger than 126 Ma record heterogeneous cooling ages on the detachment fault [20,21]. These ages have been interpreted as a record of late-stage tectonic activity [19,24,25]. However, Zhang et al. [24] noted that low initial $^{40}Ar/^{36}Ar$ values might reflect argon loss, and there is little other evidence of thermal overprinting events.

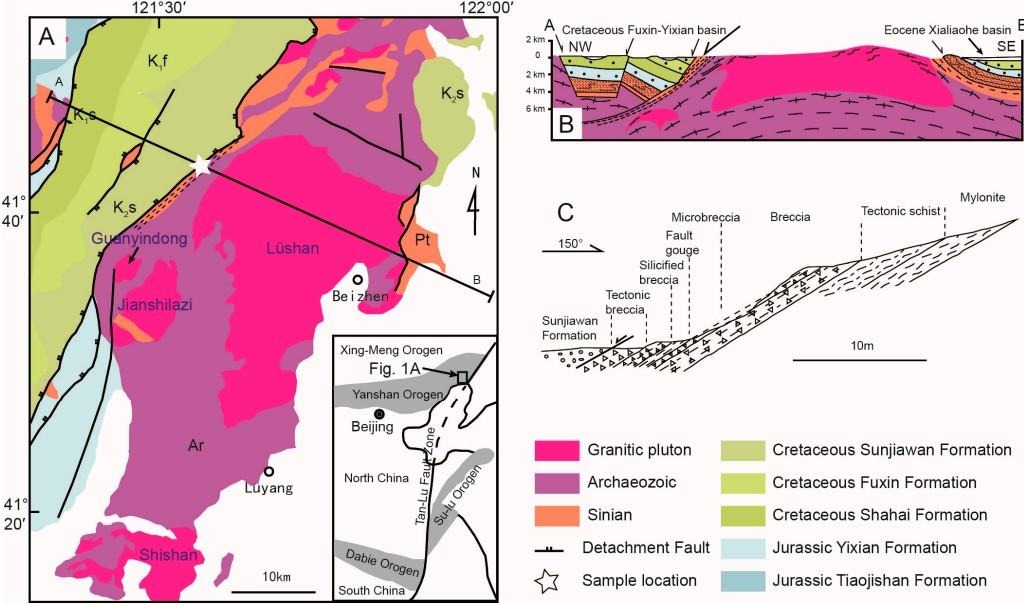

**Figure 1.** (**A**) Geological map of the Yiwulüshan area. (**B**) Cross-section perpendicular to the Waziyü detachment fault on line A–B on Figure 1A (modified from Lin et al., [20]). (**C**) Cross-section of the Waziyü detachment fault (modified from Li et al. [26]).

## 3. Sample Collection and Analytical Techniques

### 3.1. Sample Collection

A sample containing feldspar augen (CR99) was collected from quartz–mica schist within the Waziyü detachment fault to the northwest of the Yiwulüshan MCC at 41°42.532′ N, 121°34.988′ E (Figure 1A). Brittle fractures formed within the feldspar during deformation (Figure 2A). The feldspar augen are surrounded by layers of biotite and quartz–feldspar, and the feldspar phenocrysts are slightly pink in color. Scanning electron microscope (SEM) imaging (Figure 2G,H) and electron probe microanalysis (EPMA) showed that the biotite layers include phengitic white mica, quartz, plagioclase, epidote, apatite, and zircon (Table 1 and Figure 2) with grain sizes of ~20–100 μm. The imaging and the analysis were performed at the Institute of Geology and Geophysics, Chinese Academy of Sciences (IGGCAS), Beijing.

Epidote is the main alteration product of biotite. Most epidote grains are xenomorphic to subhedral with rare euhedral grains (Figure 2C,E,F). Thin epidote layers occur between the biotite layers; these connect the epidote grains (Figure 2D,E) and indicate that the biotite cleavage provided a channel for fluid flow during epidote formation. The elements that form the epidote were derived from biotite dissolution. Biotite is also chloritized, but this is less common than epidotization (Figure 2D). Rare intergrowths of phengitic white mica and biotite occur adjacent to feldspar (Figure 2F).

**Table 1.** Electron microprobe compositional analyses of CR99 epidote, phengitic white mica, biotite, chloritized biotite and K-feldspar (wt%). Analyses were performed on a JEOLJXA-8100 electron microprobe with a Wavelength Dispersive Spectrometry (WDS)/Energy-Dispersive Spectrometry (EDS) combined micro-analyzer. n/a: not applicable.

| Mineral | Na$_2$O | F | BaO | Cl | MgO | SiO$_2$ | TiO$_2$ | FeO | Al$_2$O$_3$ | Cr$_2$O$_3$ | K$_2$O | MnO | CaO | Total |
|---|---|---|---|---|---|---|---|---|---|---|---|---|---|---|
| Epidote | 0.03 | n/a | n/a | 0.01 | n/a | 38.64 | 0.08 | 10.98 | 23.47 | 0.07 | 0.01 | 0.07 | 22.66 | 96.00 |
|  | n/a | n/a | 0.03 | 0.01 | 0.06 | 38.53 | 0.09 | 11.77 | 23.04 | 0.03 | n/a | 0.10 | 22.61 | 96.26 |
|  | n/a | n/a | n/a | n/a | 0.01 | 38.16 | 0.12 | 11.11 | 23.39 | 0.05 | n/a | 0.16 | 22.78 | 95.79 |
| Phengitic white mica | 0.41 | n/a | 0.63 | 0.27 | 2.82 | 48.24 | 0.50 | 4.57 | 24.48 | 0.39 | 10.12 | 0.03 | 0.07 | 92.46 |
|  | 0.13 | n/a | 0.48 | n/a | 3.21 | 49.24 | 0.53 | 4.74 | 25.00 | 0.04 | 9.90 | 0.05 | 0.03 | 93.35 |
|  | 0.26 | n/a | 0.34 | n/a | 2.99 | 49.05 | 0.04 | 3.27 | 25.67 | 0.68 | 10.25 | 0.01 | 0.08 | 92.64 |
| Biotite | 0.07 | n/a | 0.06 | n/a | 12.67 | 38.26 | 1.70 | 16.78 | 15.79 | 0.09 | 9.69 | 0.27 | 0.05 | 95.41 |
|  | 0.06 | n/a | 0.08 | n/a | 12.50 | 38.55 | 1.58 | 15.45 | 16.10 | n/a | 9.27 | 0.26 | 0.09 | 93.92 |
| Chloritized biotite | 0.14 | n/a | 0.05 | 0.01 | 11.17 | 35.59 | 1.59 | 14.68 | 14.18 | 0.30 | 7.35 | 0.20 | 1.74 | 86.99 |
|  | 0.18 | n/a | 0.04 | 0.01 | 12.88 | 32.70 | 0.53 | 14.82 | 16.24 | 0.43 | 2.42 | 0.25 | 2.00 | 82.49 |
| K-feldspar | 0.93 | 0.01 | 1.40 | 0.02 | 0.03 | 64.13 | 0.05 | 0.10 | 18.56 | n/a | 14.57 | n/a | 0.01 | 99.79 |
|  | 0.92 | 0.09 | 1.30 | n/a | n/a | 64.59 | 0.05 | n/a | 18.51 | 0.04 | 14.57 | 0.05 | n/a | 100.09 |

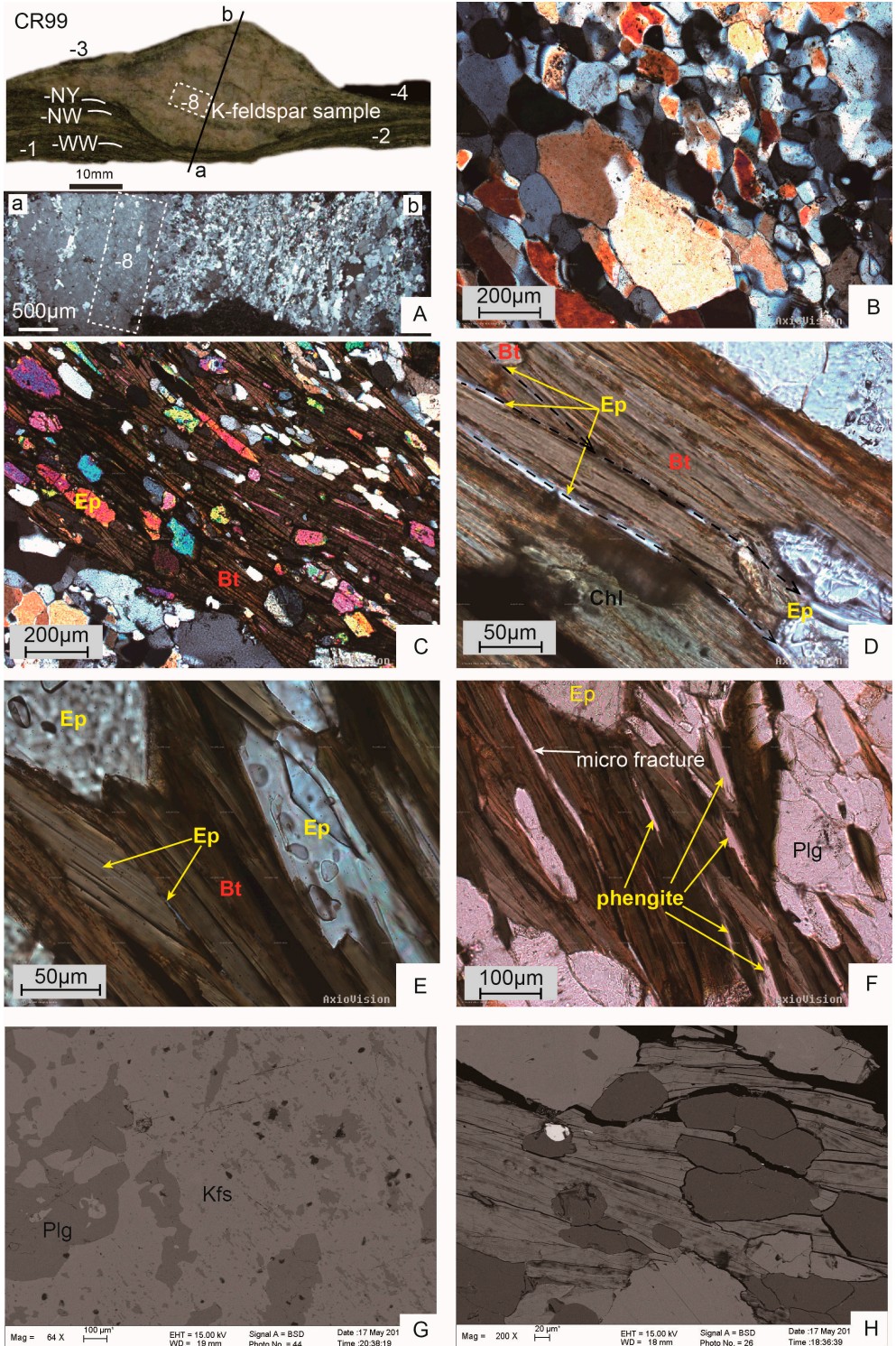

**Figure 2.** (**A**) Feldspar augen and sample preparation; the lower image shows the brittle deformation and the integrity of the sample site for sample CR99-8. (**B**) Brittle fractures in feldspar. (**C**) Biotite layers containing epidote, quartz, and plagioclase, with grain sizes of <250 μm, surrounding K-feldspar augen. (**D**) Chloritized biotite and rod-like epidote connected by an epidote grain. These textures indicate that epidote growth was related to biotite dissolution. (**E**) Rod-like epidote on the biotite cleavage. (**F**) Phengitic white mica intergrown with Biotite. (**G**) Scanning electron microscopy image of feldspar augen composed mainly of K-feldspar. (**H**) Epidote, plagioclase, quartz, and zircon separating layers of altered biotite. Ep = epidote; Bio = biotite; Chl = chlorite; Kfs = K-feldspar; Plg = plagioclase.

## 3.2. $^{40}Ar/^{39}Ar$ Dating

One K-feldspar sample, nine biotite samples, and three grains of phengite were prepared for $^{40}Ar/^{39}Ar$ dating. The K-feldspar was crushed to ~180–250 μm, and monomineralic grains were selected using a microscope and numbered as CR99-8 (Figure 2A). The K-feldspar was dated by step-heating.

Biotite samples CR99-1, -2, -3, and -4 with grain sizes of 180–250 μm were collected around the feldspar augen. Three biotite samples with grain sizes of 150–180 μm were collected at the position of CR99-1 from different biotite layers separated by felsic layers. From the inner to the outer biotite layers, these are CR99NY, CR99NW, and CR99WW CR99M and CR99M150 are mixtures of CR99NY, CR99NW, and CR99WW with grain sizes of 120–150 and 100–120 μm, respectively. All samples were analyzed by a traditional step-heating method. A split of CR99NW was analyzed by detailed step-heating, single grain fusion, and in situ laser ablation methods, and a split of CR99M150 was analyzed by detailed step-heating only. The sample positions and names are shown in Figure 2A. Three grains of phengitic white mica were separated from CR99NW and dated by in situ laser ablation methods.

The K-feldspar and the biotite were cleaned three times in deionized water and acetone. After drying at 60 °C on a hot plate, 3–11 mg biotite and 15 mg K-feldspar were packed into high purity aluminum foil. The diameter of aluminum wafers was 4.0 mm, and the thickness was ~0.5–0.75 mm. The samples were packed into a silica tube with standard YBCs sanidine (Yabachi sanidine) flux monitors and sealed under vacuum [27]. The samples were irradiated on the B4 channel of the 49-2 nuclear reactor at the China Institute of Atomic Energy, China. Cadmium foil that was 0.5 mm thick was used as a shield against thermal neutrons. The samples were irradiated for 24 h, the instantaneous flux of neutrons was $2.65 \times 10^{13}$ n cm$^{-2}$ s$^{-1}$, and the fluence was $2.28 \times 10^{18}$ n cm$^{-2}$. The vertical neutron flux gradient was about 0.4%/mm during irradiation.

The age measurements were performed at the $^{40}Ar/^{39}Ar$ and U–Th–He laboratory at IGGCAS. Two mass spectrometers were used: a Micromass UK Limited MM5400 with a double vacuum resistance furnace was used for step-heating, and a Nu instruments Noblesse spectrometer equipped with a New Wave MIR10-50 CO$_2$ laser and Analyte G2 193 nm laser from CETAC Technologies Inc. (Omaha, NE, USA) was used for total fusion and in situ laser ablation measurements.

The samples analyzed by traditional step-heating were degassed at 650 or 700 °C for 30 min, then heated from 700 or 750 °C to 1400 °C at 30–50 °C temperature intervals. For the detailed step-heating, the biotite samples were degassed at 300 °C for 30 min, then heated from 350 or 360 °C to 1300 °C at 20–50 °C temperature intervals. The K-feldspar samples were degassed at 400 °C for 30 min, and step heating schedule was implemented from 450 to 1500 °C at 50 °C intervals. Paired isothermal steps were used between 450 to 875 °C. The duration of heating at the temperature 1100 °C was increased from 15 to 240 min to maximize gas release prior to breakdown of the feldspar structure.

For the single grain total fusion analyses, single grains of biotite were transferred from the aluminum packet to a high purity oxygen-free copper plate, 4 cm in diameter, with 78 holes. The holes were 2 mm in diameter and depth. The plate was transferred to the laser chamber, which was sealed and degassed at 140 °C for 4 days. The New Wave MIR-50 CO$_2$ laser was used for sample fusion. The sample was degassed for 30 s at 0.08 W laser power, and ~1.68 W laser power was used for total fusion. The gases were purified for 5 min before introduction into the Noblesse mass spectrometer. Fusion and gas purification were performed simultaneously.

For in situ laser ablation, phengitic white mica and biotite grains were transferred from the aluminum packet to an indium plate and planished on a smoothing board. The copper and the indium plates were placed inside the laser chamber and degassed for 4 days at 160 °C. An Analyte G2 193 nm laser was used for ablation; the spot size was 50–110 μm, the laser frequency was 30 Hz, and the laser energy was 7.1 mJ. A pre-ablation procedure was also performed to minimize the effects of adsorbed air.

The gas was purified by two SAES NP10 Zr–Al getters for ~20 min before introduction to the MM5400 mass spectrometer; the gas purification time for the Noblesse mass spectrometer was 10 min. The data were acquired in peak jumping mode, and all data were corrected for blanks and mass discrimination. Interference of $^{39}$Ar on $^{37}$Ar was corrected assuming that $[^{40}Ar/^{39}Ar]_K = 3.3 \times 10^{-4}$,

$[^{36}Ar/^{37}Ar]_{Ca} = 2.69 \times 10^{-4}$, and $[^{36}Ar/^{37}Ar]_{Ca} = 8.52 \times 10^{-4}$. The K decay constant used was $5.543 \pm 0.010 \times 10^{-10}$ a$^{-1}$ [28]. Step-heating ages were calculated using the ArArCALC2.4 software (2.40, Scripps Institution of Oceanography, University of California, San Diego, CA, USA). And ArArCALC2.5 was used for laser fusion and ablation data [29]. Raw data are given in Supplementary Table S1 (step-heating results) and Table S2 (laser fusion and ablation data) [30]. Ages uncertainties are quoted at 2t level.

## 4. Results

### 4.1. K-Feldspar Age

Sample CR99-8 produced a slightly staircase-shaped age spectra (Figure 3) with apparent ages of 113.4 ± 1.3 to 119.5 ± 1.1 Ma and a more pronounced saw-tooth age spectrum from 450 to 600 °C that yields apparent ages of ~399–124 Ma. Two plateaus in the medium and the higher temperature ranges yielded ages of 113 and 117 Ma, respectively, with a cumulative $^{39}$Ar release of 27% and 11%, respectively. The maximum age of 119.5 ± 1.1 Ma was measured on the penultimate step adjacent to the 117 Ma plateau. The age of the final step was the youngest age, at 105.5 ± 2.6 Ma, which is associated with <1% of the cumulative $^{39}$Ar release. The saw-tooth pattern at low temperatures indicates that this part of the spectrum was affected by excess $^{40}$Ar*.

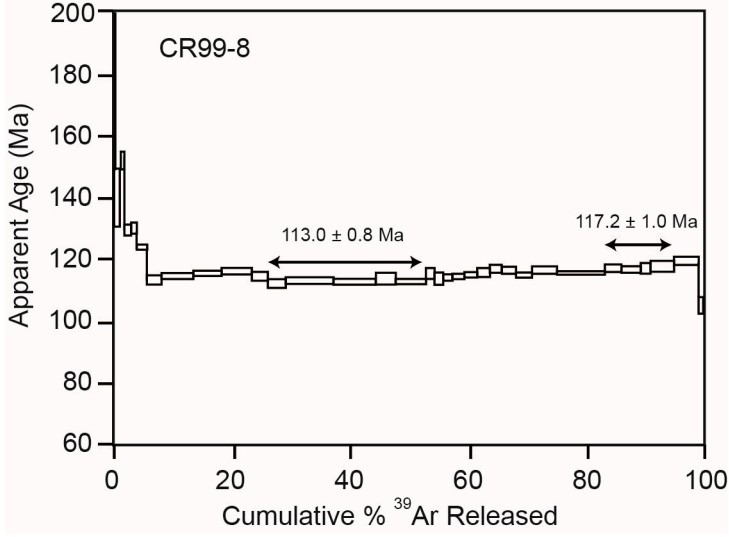

**Figure 3.** $^{40}$Ar/$^{39}$Ar results for K-feldspar sample CR99-8.

### 4.2. Step-Heating Biotite Data

The traditional step-heating analyses produced similar age spectra (Table 2). The lowest temperatures yielded relatively young ages for all spectra, which increased to a higher apparent age, followed by a dip at medium temperatures and increasing apparent ages with increasing temperature. This pattern is similar to that described by Lo et al. [7]. The trends in apparent age are related to the K/Ca ratios of the samples; samples with the highest K/Ca ratios yielded the highest apparent ages and vice versa. Therefore, the younger apparent ages might reflect biotite alteration.

Four samples produced plateau ages. The steps that formed the plateaus were located within the medium-temperature range and included, or were adjacent to, the youngest age step. The plateau and the inverse isochron ages of CR99-1 and -4 were similar: 119.0 ± 1.9 and 119.2 ± 17.3 Ma for CR99-1 and 121.0 ± 1.4 and 122.2 ± 4.6 Ma for CR99-4, respectively, with initial $^{40}$Ar/$^{36}$Ar ratios close to the atmospheric value of 295.5. The plateau ages of CR99WW and CR99M were younger, at 92.5 ± 2.1 and 105.6 ± 0.9 Ma, respectively, but the initial $^{40}$Ar/$^{36}$Ar ratios derived from inverse isochron plots

(272.1 ± 64.2 and 267.1 ± 71.7, respectively) indicated argon loss. Samples CR99NW, CR99NY, CR99M, and CR99M150 yielded similar young ages of 105.6 ± 1 Ma for the medium-temperature range.

**Table 2.** Summary of traditional and detailed step-heating data. Superscript D refers to degassing and S denotes the starting temperature of the age analysis. Superscript a refers to plateau age calculated from at least two steps. T.F. = total fusion; Flat I = first older age plateau; Flat II = second older age plateau. All errors are reported at 2σ level.

| Sample | Size (μm) | Tem. (°C) | Age (Ma) | | | | | | | $(^{40}Ar/^{36}Ar)_i$ |
|---|---|---|---|---|---|---|---|---|---|---|
| | | | Initial | Flat I | Dip | Flat II | T.F. | Plateau | Isochron | |
| CR99-1 | | | 112.3 ± 2.1 | 129.5 ± 1.2 | 108.8 ± 2.0 | 137.9 ± 1.9 | 121.6 ± 0.7 | 119.0 ± 1.9 | 119.2 ± 17.3 | 294.8 ± 80.3 |
| CR99-2 | 180–250 | 650$^D$ | 107.6 ± 2.3 | 129.0 ± 1.2 | 95.2 ± 3.5 | 145.5 ± 1.3 | 118.7 ± 0.8 | n/a | n/a | |
| CR99-3 | | 700$^S$ | 124.0 ± 1.0 | 140.3 ± 0.7 | 121.1 ± 1.0 | 138.9 ± 1.5 | 131.7 ± 0.7 | n/a | n/a | |
| CR99-4 | | | 115.9 ± 2.2 | 130.8 ± 1.3 | 121.0 ± 1.4$^a$ | 132.5 ± 2.8$^a$ | 124.2 ± 0.7 | 121.0 ± 1.4 | 122.2 ± 4.6 | 288.1 ± 26.4 |
| CR99WW | | | 102.7 ± 2.3 | | 89.0 ± 2.3 | 112.8 ± 2.7$^a$ | 95.4 ± 1.0 | 92.5 ± 2.1 | 98.0 ± 15.1 | 272.1 ± 64.2 |
| CR99NW | 150–180 | 700$^D$ | 116.8 ± 1.1 | | 122.5 ± 1.0 | 105.6 ± 1.1$^a$ | 122.3 ± 1.9 | 112.9 ± 0.6 | n/a | n/a | |
| CR99NY | | 750$^S$ | 123.6 ± 1.5 | 127.9 ± 1.2 | 105.8 ± 1.0 | 126.5 ± 1.3$^a$ | 117.6 ± 0.7 | n/a | n/a | |
| CR99M | 120–150 | | 118.3 ± 1.4$^a$ | | 105.6 ± 0.9$^a$ | 122.1 ± 1.4 | 109.0 ± 0.8 | 105.6 ± 0.9 | 109.1 ± 8.9 | 267.1 ± 71.7 |
| CR99M150 | 100–120 | | 118.8 ± 1.4 | 123.4 ± 0.7 | 105.5 ± 0.7 | 130.9 ± 1.3 | 115.3 ± 0.6 | n/a | n/a | |
| CR99NW | | 350$^S$ | 4.6 ± 13.1 | 119.4 ± 1.3 | 104.4 ± 0.7 | 120.8 ± 1.6 | 90.8 ± 0.8 | n/a | n/a | |
| CR99M150 | | 360$^S$ | 7.8 ± 7.4 | 121.0 ± 1.0 | 104.5 ± 0.6 | 123.2 ± 3.3 | 98.1 ± 0.7 | n/a | n/a | |
| CR99-8 | 180–250 | 450$^S$ | | 113.0 ± 0.8 | | 117.2 ± 1.0 | 117.3 ± 0.7 | | | |

Detailed step-heating of CR99NW and CR99M150 produced similar age spectra, which differed from the traditional step-heating results at low temperatures. Approximately 0.2% of the total $^{39}$Ar was released by CR99NW between 350 and 440 °C, which yielded ages of 78–40 Ma with large uncertainties. Subsequently, the apparent age increased from 5.5 to 120 Ma, decreased to 105 Ma, and increased to ~120 Ma (Figure 4A). Approximately 2% of the $^{39}$Ar was released from CR99M150 at low temperatures and yielded ages of 130–23 Ma. The ages then increased from 8 to 121 Ma, decreased to 104 Ma, and increased to 123 Ma. Notably, the $^{39}$Ar released at 440–750 °C was ~50% of the total, indicating that the traditional biotite step-heating data omitted the low-temperature data and argon diffusion information and that the total fusion ages were overestimates.

Sample degassing during detailed incremental heating was characterized by early $^{36}$Ar release followed by release of $^{39}$Ar$_K$ and $^{40}$Ar*. Degassing of $^{36}$Ar increased above 380 °C and peaked sharply at 400–700 °C. The $^{39}$Ar$_K$ and the $^{40}$Ar* degassing peaks were located at 440–700 °C and 800–1000 °C, respectively (Figure 4B). Two peaks of argon release for $^{39}$Ar$_K$ and $^{40}$Ar* are common in biotite analyses [7]. Lo et al. [7] documented a shift toward higher temperatures for the second peak for samples with low Fe/(Fe + Mg) values, but the second peak for our samples (Fe/(Fe + Mg) ~ 0.424) occurred at a similar temperature to the second peak in the Lo et al. study (Fe/(Fe + Mg) ~ 0.644). Moreover, the fraction of $^{39}$Ar$_K$ released was higher than the fraction of $^{40}$Ar* released at temperatures <570 °C and <540 °C for CR99NW and CR99M150, respectively. At temperatures higher than these, the proportion of $^{40}$Ar* was greater than that of $^{39}$Ar*. These temperatures might correspond to the temperature of delamination associated with biotite dehydroxylation [7].

The traditional step-heating method omits a substantial proportion of the signal and overestimates the total fusion age, but the traditional and the detailed step-heating apparent ages for CR99NW and CR99M150 are comparable for corresponding temperature intervals. In the traditional step-heating results, the apparent ages decreased with decreasing grain size. This trend does not reflect a relationship between grain size and sample age, because the apparent ages are much older than the age of coexisting K-feldspar. Instead, it might reflect chloritization effects and the presence of fluid inclusions within minerals such as epidote, quartz, and plagioclase based on petrographic observations. In this case, the older ages can be attributed to $^{39}$Ar$_K$ recoil and excess argon within these minerals and their presence in the biotite samples. These minerals are excluded more easily during grain selection when the grain size is 150–180 μm, and the ages for this grain size fraction are more reproducible. Furthermore, given that $^{40}$Ar* gradients are common within biotite grains, sub-optimal degassing releases a higher proportion of $^{39}$Ar$_K$ than $^{40}$Ar*, such that traditional step-heating methods and integrated age calculations yield high apparent ages.

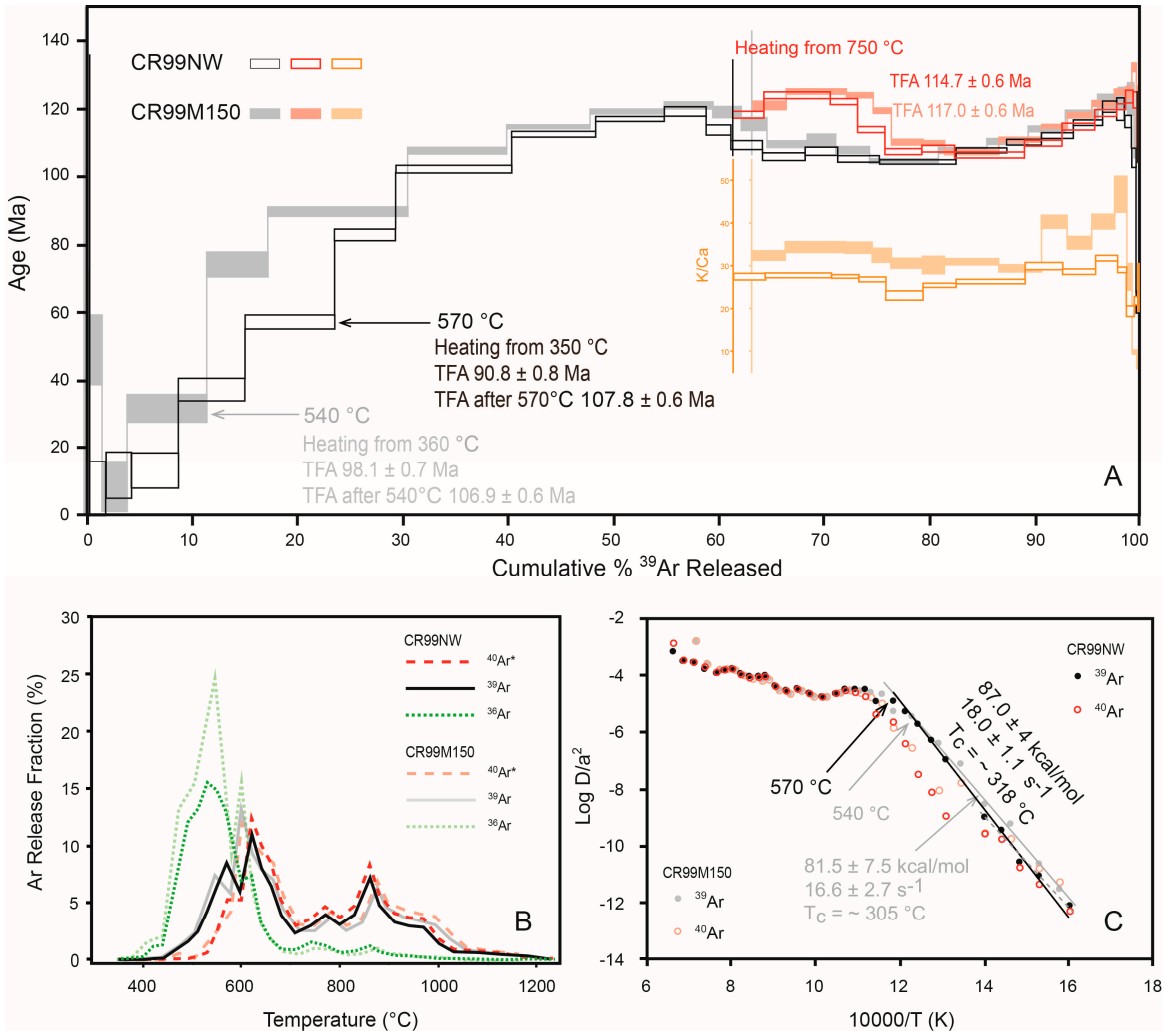

**Figure 4.** Results of detailed step-heating of CR99NW and CR99M150. (**A**) Age spectra. The red age spectra were produced by traditional step-heating from 750 °C. Compared with the detailed step-heating results, the first age peak of these two samples aws shifted to higher temperature, but the apparent ages were similar. The K/Ca spectrum of the traditional step-heating data is shown in orange below the age spectrum. TFA = total fusion age. (**B**) Argon release pattern: the change in the $^{40}$Ar* and the $^{39}$Ar pattern at ~570–600 °C (~540–570 °C for CR99M150) might record breakdown of the biotite structure. Older apparent peak ages were produced consistently towards the upper temperature end of the release peak. (**C**) Arrhenius diagram: closure temperature ($T_C$) was calculated assuming a cooling rate of 100 °C Myr$^{-1}$. The regression was terminated by the change in gradient at 570 °C for CR99NW and 540 °C for CR99M150.

## 4.3. Total Fusion Biotite Ages

We analyzed 160 grains of CR99NW. Most of the grains were too small to obtain a satisfactory signal/blank ratio (>5:1), and the proportion of atmospheric argon was high, and thus the errors were large. Consequently, only 81 ages with errors <10%, or >20% radiogenic $^{40}$Ar were considered further. The ages ranged from 77.4 ± 10.6 to 188.0 ± 3.4 Ma, and the oldest age was 327.9 ± 5 Ma. The ages formed a continuum from 77 to 121 Ma, with a peak at ~105 Ma (Figure 5).

We did not measure the size of each grain, and thus the $^{39}$Ar$_K$ signal was used as a proxy for grain size. A plot of the biotite single grain total fusion age versus the $^{39}$Ar$_K$ signal showed that the degree of age variability decreased with increasing $^{39}$Ar$_K$ signal, indicating that the total fusion ages are controlled by grain size or degree of alteration and the fraction of argon loss (Figure 5).

The total fusion ages of single grains provide a valuable opportunity to test the homogeneity of the samples. A heterogeneous age distribution might reflect differential partial resetting of the argon system or the presence of excess argon. We infer that grains with apparent ages of >120 Ma contained excess $^{40}Ar^*$ ($^{40}Ar_E$) and that grains with apparent ages of <110 Ma underwent partial argon loss.

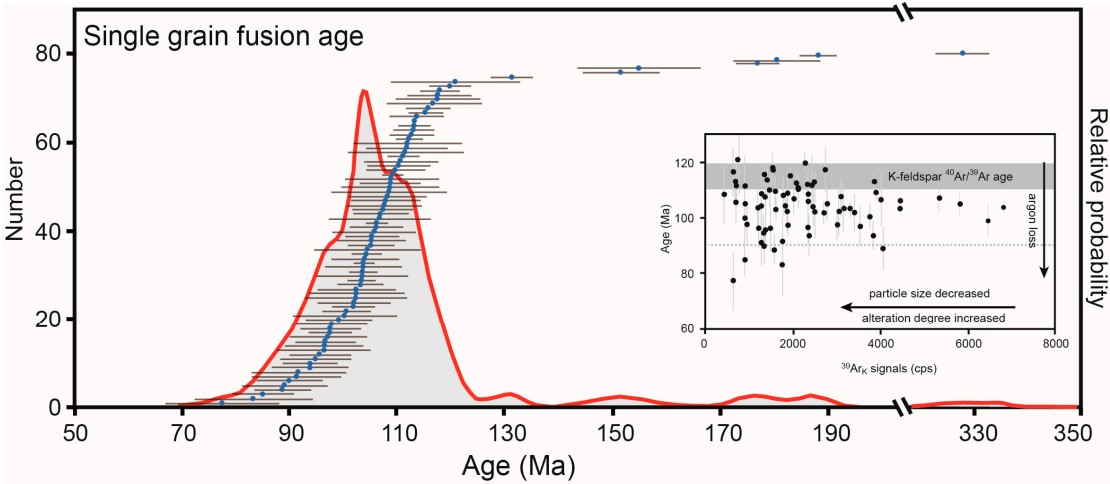

**Figure 5.** Single grain fusion age distributions (blue circles with gray bars) and relative probability density plots (red curve). Inset shows total fusion ages versus $^{39}Ar_K$, indicating that a high $^{39}Ar_K$ signal is associated with ages close to ~105 Ma and the peak of the red curve. Ages of >122 Ma are not shown in the inset, but the $^{39}Ar_K$ signals associated with these ages are 1000–4000 cps.

### 4.4. In Situ Laser Ablation Ages

Several grains of CR99NW were analyzed with an in situ laser ablation technique. However, high-precision results were not obtained for most of the grains because of their limited thickness and the presence of large amounts of atmospheric contamination, and thus the ages obtained from different spots on single grains were within error. Two grains yielded consistent ages: three spots on one grain record an age of 105.7 Ma, and another grain yielded an age of 96.5 Ma. Only two grains showed age trends within the grains. Ages of 41.6 ± 7.3, 64.4 ± 9.3, and 60.2 ± 12.1 Ma were obtained from one grain, and 64.3 ± 7.7, 61.0 ± 8.9, 72.8 ± 10.6, and 80.5 ± 10.2 Ma were obtained from another grain (Figure 6).

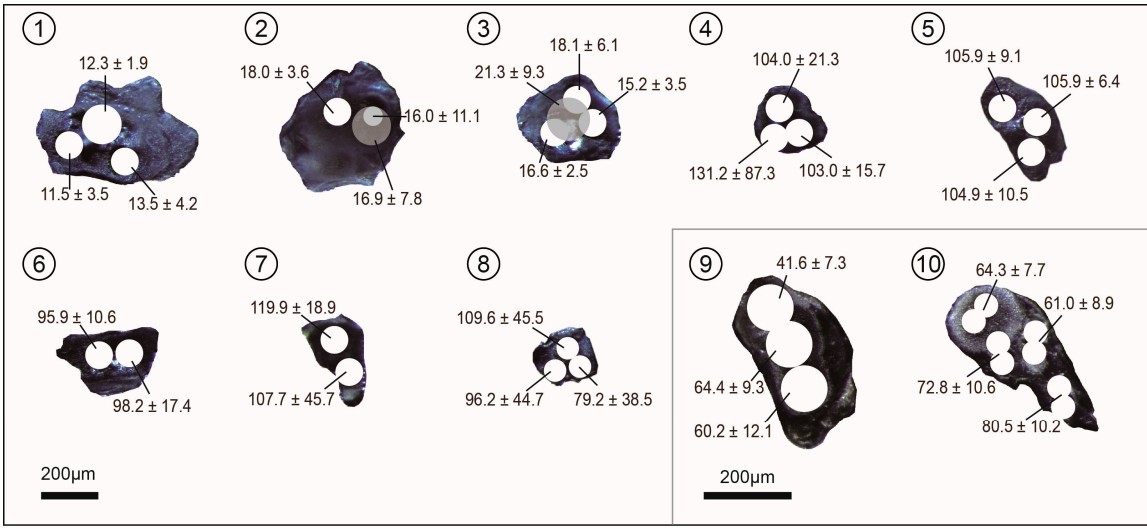

**Figure 6.** In situ laser ablation ages of biotite (4–10) and phengitic white mica (1–3).

Three phengitic white mica grains separated from this sample yielded two different young ages. The spot ages from each grain were indistinguishable within error. The weighted-mean ages of these three grains were 12.3 ± 1.5, 17.7 ± 3.1, and 16.6 ± 1.9 Ma (Figure 6).

## 5. Discussion

### 5.1. The Effect of Epidotization on Biotite Argon Systematics

Biotite within CR99 is extensively epidotized. The presence of epidote records metamorphism and hydrothermal activity at 250–400 °C and 1–2 kbar [31]. Eggleton and Banfield [32] reported that biotite from a granite reacted to form epidote, titanite, and chlorite under hydrothermal conditions at 330–340 °C. This temperature range is within the range of biotite closure temperatures (280–345 °C) and corresponds to an experimental activation energy of 47.0 ± 2 kcal mol$^{-1}$, a frequency factor of $0.077^{+0.21}_{-0.06}$ cm$^2$ s$^{-1}$, and an effective diffusion radius of about 150 µm [33]. It is possible to quantify argon loss during heating, but only if volume diffusion is the main process that controls argon loss.

Biotite $^{40}$Ar/$^{39}$Ar dating is based on assumptions that describe the diffusion of argon in biotite and, as such, many studies have focused on this process. If diffusion is thermally activated, then the diffusion coefficient can be estimated using a simple Arrhenius Law:

$$D = D_0 \exp(-E/RT), \tag{1}$$

where R is the gas constant, $T$ is the absolute temperature, $E$ is the activation energy, $D$ is the diffusion coefficient, and $D_0$ is the diffusion coefficient at very high values of $T$, also known as pre-exponential frequency factor. If diffusion is controlled by a single mechanism, then the data plot on a line on an Arrhenius plot and $E$ and $D_0$ can be calculated from the slope and the intercept of a linear regression of the data. These two parameters are known as the diffusivity kinetic parameters and control argon diffusion within minerals.

With the exception of the older ages produced at <490 °C, CR99NW and CR99M150 yielded a staircase-shaped pattern of young ages at 500–600 °C. These parts of the spectra were associated with the release of 29.4% and 30.6% of the cumulative $^{39}$Ar$_K$, respectively, in contrast to the steps at <500 °C where only 0.2% and 1.7% of the cumulative $^{39}$Ar$_K$ was released, respectively. The corresponding log($D/a^2$) values of $^{39}$Ar$_K$ determined from an Arrhenius plot, where $a$ is the effective diffusion radius, were larger than $^{40}$Ar*, indicating that $^{39}$Ar$_K$ was released faster than $^{40}$Ar* during these temperature steps (Figure 4C). If argon mobility was controlled by volume diffusion, then there must have been an $^{40}$Ar* concentration gradient within the biotite grains and depletion in $^{40}$Ar* at the grain rim [8]. If argon diffusion was controlled by short-circuit diffusion, then the young ages did not have any geological meaning. Crystal defects provided the fast diffusion pathways for short-circuit diffusion, and $^{39}$Ar$_K$ might have been implanted in these sites by recoil during neutron irradiation. Excess $^{40}$Ar* could also be trapped in defects. During in vacuo heating, $^{39}$Ar$_K$ and $^{40}$Ar* located in defects were released early, and excess $^{40}$Ar* or recoil-implanted $^{39}$Ar$_K$ could produce anomalously old or young ages.

The older ages produced by these two samples at temperatures of <490 °C are inferred to reflect excess $^{40}$Ar* trapped by crystal defects or fluid inclusions. The proportion of $^{40}$Ar* released during these steps was higher than the proportion of $^{40}$Ar* released at the subsequent step, while the released proportion of $^{39}$Ar$_K$ increased gradually as temperature increased. On an Arrhenius plot, there was a slightly different gradient for the temperature steps between 500 and 600 °C and those below 490 °C, indicating that there might have been small differences in the rate of $^{39}$Ar$_K$ release for CR99NW. However, the gradients were within error of each other. A linear regression could be applied to the data corresponding to the first seven temperature steps for CR99M150, indicating that Ar diffusion within these samples at low temperatures (<570 °C) was dominated by a single mechanism, which might have been volume or short-circuit diffusion. However, short-circuit diffusion is typically associated with low activation energies [34]. Diffusion rate information can only be derived if the mineral phase is stable

throughout the heating process, thus there are few examples of the use of in vacuo heating results to constrain Ar diffusion in biotite. However, some historical data produced by in vacuo heating might be applicable. We compared step-heating results that included low-temperature steps [7,33,35,36], which included data for chloritized and unaltered biotite with hydrothermal diffusion data (Figure 7). Our data yielded relatively high activation energies within the range of those obtained for normal biotite. Based on these kinetic parameters, we infer closure temperatures of ~312 °C and ~300 °C for CR99NW and CR99M150, assuming a cooling rate of 10 °C Myr$^{-1}$. This is similar or slightly lower than the closure temperature derived from hydrothermal experiments that assumed an effective diffusion length of 150 µm (~310 °C, [33,36]) and indicates that biotite layers retain their crystal structure after epidotization. These results contrast with those of Lo et al. [7] for chloritized biotite, which imply a relatively low activation energy and a closure temperature of <200 °C.

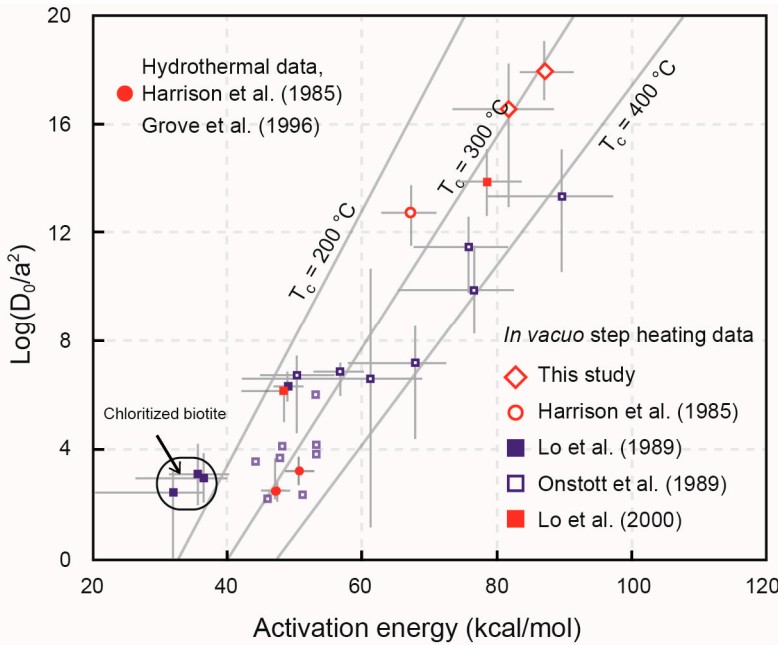

**Figure 7.** Kinetic parameters for diffusion of $^{39}$Ar in biotite, chloritized biotite, epidotized biotite, and $^{40}$Ar* in biotite. All data are derived by in vacuo step-heating except for the $^{40}$Ar* data from hydrothermal experiments. The open squares without error bars are from Onstott et al. [35] and were calculated by an interpolation of the pre-exponential frequency factor ($D_0$) and activation energy ($E$) against the Fe/(Fe + Mg) value of the biotite sample based on the diffusion data of Harrison et al. [33] and Giletti [37]. The lines correspond to closure temperatures between 200 and 400 °C, assuming cooling at 10 °C Myr$^{-1}$.

If the cooling rate is assumed to be 100 °C Myr$^{-1}$, then the diffusion parameters of Harrison et al. [33] indicate a closure temperature of 345 °C, and the biotite age is close to 119 Ma, which is comparable to the $^{40}$Ar/$^{39}$Ar age of K-feldspar. If the kinetic parameters derived from the Arrhenius diagram are applied, then the closure temperature is ~330 °C for CR99NW (Figure 4C) and ~318 °C for CR99M150, corresponding to a cooling age of ~118 Ma. The continuous increase in the ages derived from single grain total fusion $^{40}$Ar/$^{39}$Ar data ceases at ~120 Ma, and the oldest apparent ages produced by detailed step-heating are also close to ~120 Ma. Some grains record total fusion ages older than this; these might reflect the age of relict biotite where ductile deformation did not completely reset the Ar systematics or grains with excess $^{40}$Ar*.

The total fusion ages obtained by detailed step-heating of biotite (91 Ma for CR99NW and 98 Ma for CR99M150) indicate that a late thermal event caused partial argon loss. During this event, CR99NW and CR99M150 lost at least 23.5% and 17.6% of their $^{40}$Ar*, respectively, if the original age of the

biotite was 119 Ma. However, the activation energy and the diffusion factors calculated from the Arrhenius plot show that CR99NW retained more argon than CR99M150 [38], thus CR99M150 should have lost more $^{40}$Ar* and produced a younger total fusion age. This contradictory finding indicates that hydrothermal fluids were not the only control on partial $^{40}$Ar* loss and that the extent of sample alteration should also be considered. This might explain why Ma et al. [18] measured a biotite K–Ar age of 81 Ma for this area and the variable $^{40}$Ar/$^{39}$Ar ages of Zhang et al. [24] and Lin et al. [20]. The distribution of single grain total fusion ages and age gradients within single grains revealed by laser ablation dating support this conclusion. However, single grain total fusion and in situ laser ablation biotite $^{40}$Ar/$^{39}$Ar ages are interpreted as mixed ages, because differently altered biotite grains might retain different age gradients.

Typically, epidote grains are subhedral to euhedral, with rod-like epidote on the biotite cleavage. These observations indicate that epidotization occurred at various times during fault activity, and that biotite underwent multiple or extended periods of alteration in addition to the thermal overprint. The total fusion and the ablation ages cannot distinguish between single and multiple late thermal events because of the micron-scale alteration. However, these techniques provide a lower age constraint on the latest event. The single grain total fusion ages indicate an event after ~80 Ma, and the youngest spot age of 47 Ma indicates that the last stage of epidotization was after this time, because the requirement of high enough signal-to-instrumental blank limited the use of higher spatial resolution of laser ablation technique.

In thin-section, the phengitic white mica occurred adjacent to euhedral epidote. This might indicate that the latest epidotization was related to the growth of white mica. In situ laser ablation of phengite yielded ages of 18–12 Ma, similar to the youngest ages of CR99NW; the weighted-mean age of the youngest three steps was 11.6 ± 4.2 Ma (Figure 8). These analyses indicate that the low-temperature staircase age spectra of these samples might represent a real age gradient and record long-lasting hydrothermal alteration of biotite. The latest fault activity occurred at ~12 Ma based on the fluid activity recorded by epidotization and muscovitization of biotite. A geologically meaningful age for biotite alteration was recorded by the detailed step-heating results as long as argon diffusion was controlled by volume diffusion at <600 °C.

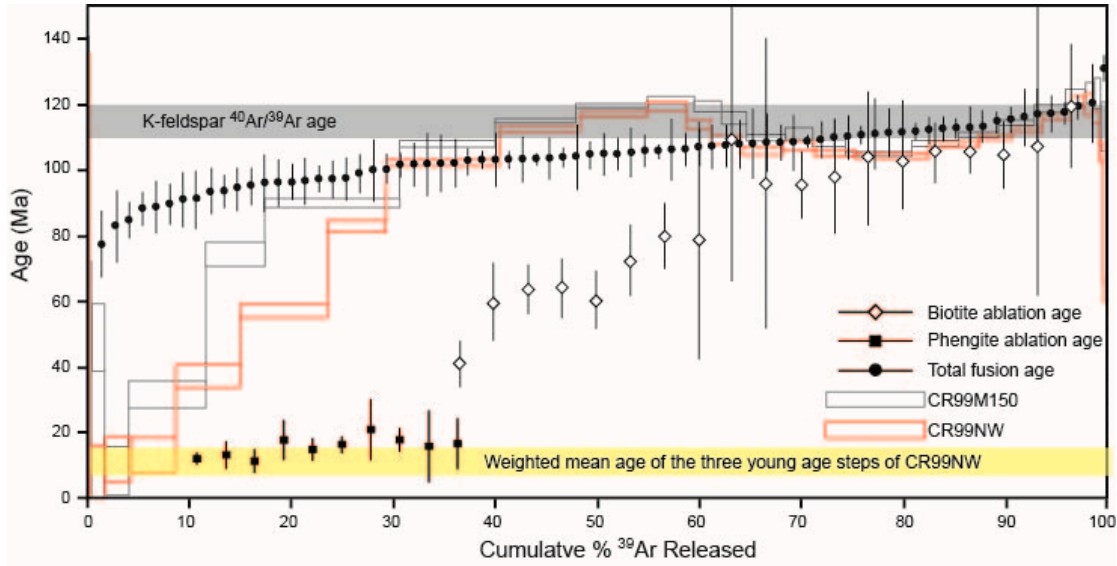

**Figure 8.** Summary of the ages of different minerals obtained by different techniques. Only the detailed step-heating technique produced ages comparable to those of phengitic white mica, which formed during biotite alteration and fault activity. Errors are plotted as 2σ.

*5.2. Implications for Interpretation of Total Fusion and In Situ Laser Ablation Ages and Disturbed Biotite $^{40}Ar/^{39}Ar$ Age Spectra*

Chloritization of biotite and $^{39}Ar_K$ recoil associated with neutron irradiation are commonly invoked to explain disturbed biotite age spectra [7,9,10]. Chloritization produces K-poor layers within biotite. During neutron irradiation, the reaction of $^{39}K(n,p)$ to $^{39}Ar$ involves energies of ~3.0 MeV, which produce recoil distances of ~0.1 μm in biotite. This means that $^{39}Ar_K$ can recoil from a layer of K-rich biotite into a layer of K-poor chlorite. Argon is less well retained by chlorite layers, thus the $^{39}Ar_K$ is released more quickly than $^{40}Ar^*$, which is located mostly within biotite layers; this process produces low $^{40}Ar^*/^{39}Ar_K$ ratios. If, on the other hand, release of $^{39}Ar_K$ is mostly from biotite, then the $^{40}Ar^*/^{39}Ar_K$ value is higher because the biotite is depleted in $^{39}Ar_K$. Loss of $^{39}Ar_K$ by recoil is typically negligible. For example, chloritized biotites have disturbed age spectra, but the total fusion ages of the bulk sample and the single grains are still close to the emplacement age [10]. In this study, the higher apparent ages inferred from step-heating and single grain total fusion dating were younger than 120 Ma, which is the oldest apparent age of K-feldspar, indicating that recoil effects were probably negligible in our samples. Furthermore, the release of $^{39}Ar_K$ matched well with the release of $^{40}Ar^*$ (Figure 4B); there was no offset between the peaks of $^{39}Ar_K$ and $^{40}Ar^*$, in contrast to the spectra of the chloritized biotite of Lo et al. [7]. We infer that the disturbed age spectra of our samples do not reflect $^{39}Ar_K$ recoil.

Chlorite layers within chloritized biotite might retain some K and $^{40}Ar^*$, and thermal effects related to fluid activity at 200–400 °C cause some loss of $^{40}Ar^*$ from biotite layers [39,40], thus it is unsurprising that the $^{40}Ar/^{39}Ar$ total fusion ages of chloritized biotite are commonly younger than cooling age. For this reason, it is difficult to obtain meaningful ages by in situ laser ablation when chloritized domains are distributed heterogeneously within biotite grains [10]. It is possible that chloritization might have affected the laser ablation $^{40}Ar/^{39}Ar$ ages of our samples. Epidotization causes some argon loss from biotite layers, but the length scale of alteration is that of the mineral grains, thus the thermal effects are limited. Grains with variable amounts of alteration record different ages with all ages younger than the cooling age. Age gradients exist in some grains, but different grains preserve different age gradients. This lack of consistency is attributed to modification of the biotite grains by epidotization. Calcium-rich hydrothermal fluids associated with fault activity infiltrated the biotite cleavage and partially dissolved the biotite. Epidote nucleated on cracks, vacancies, or defects within the biotite grains, which modified the size and the shape of the biotite grains and the original argon profile. The variability in the total fusion ages of fine-grained biotite can also be attributed to this process. The single grain total fusion and the laser ablation ages were bracketed by the range of apparent ages obtained by detailed step-heating, and the age peak of single grain total fusion ages was ~105 Ma, which is close to the younger ages obtained within the medium-temperature range by step-heating. This indicates that the young apparent ages obtained from the medium-temperature range of a disturbed age spectrum reflect the single grain ages distribution of the bulk sample.

During in vacuo heating, the crystal structure of biotite dehydroxylates and breaks down, which homogenizes age gradients [1,3]. During this process, biotite decomposition and transformation, rather than volume diffusion, control argon release, and it becomes difficult to interpret the cooling and the geological history of the sample. However, mixtures of biotite populations yield disturbed age spectra even at temperatures of >600 °C [11], and many studies have indicated that biotite is stable below this temperature [7,41]. We infer that these observations can be reconciled if argon gradients are only homogenized in specific grains. If all grains within a sample have the same age or age gradient distribution, then homogenization during in vacuo heating would produce a flat age spectrum. In contrast, if a bulk sample contains more than one population of biotite, then homogenization cannot produce a flat plateau because the argon release pattern is also related to the chemical composition of biotite grains [7,11].

We conclude that peak–valley–peak-type age spectra are produced when there is a transition in the diffusion mechanism during in vacuo heating. At low-temperature steps (<600 °C), the age

spectra reflect the age gradients within biotite grains because the biotite is thermodynamically stable. At higher temperatures, decomposition of the biotite destroys $^{40}$Ar* gradient information. However, decomposition and homogenization of $^{40}$Ar* only affect some of the grains. The chemical composition and the integrated ages of the grains might differ if the grains experience different amounts of alteration. In this case, the ages increase from 600 to 750 °C because of the chemical compositional control on diffusion, thus argon is released differently from grains of different chemical composition. However, at temperatures of >750 °C, biotite crystallinity is lost, and the chemical composition ceases to control argon diffusion. At 750–900 °C, one or two steps might produce young apparent ages or a young age plateau equal to the integrated gas age, depending on the variability of the annealing temperature within the biotite populations. Notably, the integrated gas age is not equal to the total fusion age of the bulk sample but only to the total fusion age of the sample after argon homogenization. At temperatures >900 °C, the biotite structure anneals, which reinstates the relationship between chemical composition and argon diffusion, and thus a staircase spectrum is produced at these temperatures. The absolute temperature limits on these processes are a function of the chemical composition, but the trends with changing temperature are consistent.

Other evidence supports this interpretation. X-ray diffraction (XRD) and transmission electron microscopy (TEM) studies conducted on chloritized annite heated in vacuo show that biotite dehydroxylation occurs mainly at ~600 °C, crystallinity is mostly lost by about 810 °C, and annealing at higher temperatures restores biotite crystallinity up to ~1080 °C [10,42]. High-temperature XRD analysis has revealed that biotite crystallinity can be retained to ~1000 °C [7]. A study by Kula et al. [11] showed that the youngest age of the valley part of an age spectrum is the most similar to the mean age of a mixed sample. We infer that homogenization occurs after the biotite grains lose their crystallinity, consistent with the results of XRD and TEM. Furthermore, the young apparent ages obtained for the medium-temperature range reflect the distribution of total fusion ages of biotite grains within a sample. The older ages at each side of the valley represent the original cooling age of the sample. The cooling age might be underestimated or overestimated if there are different age populations or chloritized layers within the biotite. The apparent ages prior to dehydroxylation might reflect real age gradients within biotite, but these should be treated with caution, because sample preparation artifacts or $^{39}$Ar$_K$ recoil processes can affect the apparent ages.

### 5.3. Implications for the Deformation History of the Waziyü Detachment Fault

A number of geochronological studies have attempted to constrain the exhumation history of the Yiwulüshan MCC since it was described by Ma et al. [18]. Lin et al. [20] compiled age data for the complex, including zircon U–Pb dates, and biotite, muscovite, amphibole, and K-feldspar $^{40}$Ar/$^{39}$Ar ages. Based on these data, field-based structural analysis, and laboratory work, they concluded that the Yiwulüshan massif underwent polyphase deformation. After emplacement of the Jurassic Lüshan pluton, early south-directed thrusting (D$_1$) was overprinted by later extensional deformation (D$_2$–D$_4$). Thrusting occurred at ~141 Ma, and movement on the Waziyü detachment fault occurred at ~128–126 Ma at the same time as two periods rapid cooling of the Lüshan pluton, which was revealed by K-feldspar $^{40}$Ar/$^{39}$Ar dating. Young ages of 116–97 Ma are recorded by samples from the Waziyü detachment fault, and these ages are interpreted as a record of heterogeneous cooling. Wang et al. [21] used the pre- or syn-deformation mylonitized intrusions and post-kinematic unmylonitized intrusions of the MCC to constrain the timing of middle-lower crustal ductile extension and inferred that high-temperature deformation of the Yiwulüshan massif occurred at ~152–126 Ma and that $^{40}$Ar/$^{39}$Ar ages of 127–116 Ma record uplift of the MCC.

Zhang et al. [24] documented argon loss from biotite but provided little other evidence of a late-stage thermal overprint that affected the fault. However, processes related to deformation after exhumation of the shear zone to its current shallow crustal level were not considered in this study. At this time, temperature was not the main control on argon diffusion within biotite and muscovite. Instead, dissolution and precipitation reactions, re- and neo-crystallization, and alteration related to the

circulation of hydrothermal fluids associated with fault activity caused partial or total argon loss from minerals and geologically meaningless $^{40}$Ar/$^{39}$Ar ages. Lin et al. [20] show that the Waziyü detachment fault was overprinted by brittle deformation associated with formation of the Fuxin–Yixian Basin.

Plagioclase and K-feldspar are stronger and more resistant to ductile deformation than biotite under greenschist-facies conditions, and thus their $^{40}$Ar/$^{39}$Ar ages should be more robust than those of biotite. K-feldspar is a useful thermochronometer because it can constrain the cooling history of a rock between 350 and 150 °C [43]. Here, we used K-feldspar augen to constrain the timing of deformation on the Waziyü detachment fault. The apparent ages ranged from 120 to 113 Ma. Saw-tooth age spectra were produced at low temperatures (450–600 °C), indicating that the K-feldspar interacted with fluids [44]. However, these ages were older than the apparent ages obtained for higher temperature analysis, even after correction for excess $^{40}$Ar. The $^{40}$Ar* signal at low temperatures was derived from the margins of the grains, indicating that the excess $^{40}$Ar was not solely trapped within fluid inclusions but was also present within the feldspar crystal. This might relate to feldspar recovery or dynamic recrystallization during periods of high strain related to the ductile deformation. A cooling rate of 50 °C Myr$^{-1}$ was calculated from the range of K-feldspar closure temperatures and the 4 Myr difference in ages between the maximum and the minimum $^{40}$Ar/$^{39}$Ar plateau ages. This indicates that the detachment fault cooled rapidly at 120–113 Ma. The age range is comparable with the age of Fuxin Basin subsidence and sedimentation of the Shahai and the Fuxin formations (122–110 Ma; [18]). Therefore, the detachment fault was overprinted by the normal fault movement that formed the graben basin (D4 of Lin et al., [20]).

Three grains of phengitic white mica had different spot ages (18–12 Ma), which is interpreted to reflect feldspar retrogression during interaction with a late-stage hydrothermal fluid and fault movement. There are no known tectonic events of these ages, but Cenozoic basaltic eruptions occurred within the North China Craton at this time (e.g., the Shanwang basalts (21–17 Ma; [45]) and Qixia basalts (12 Ma; [46])). Fertile peridotite xenoliths within these basalts record lithospheric thickening during the Neogene–Quaternary in response to cooling of upwelling asthenosphere [47]. These basalts erupted adjacent to the Tanlu Fault and the Yiwulüshan MCC, and thus these ages are inferred to record fault movement related to basalt eruption.

## 6. Conclusions

(1) Biotite alteration associated with hydrothermal fluid activity during fault movement causes argon loss. Partial resetting of the argon systematics and the intra-grain variations in the amount of alteration produces variable single grain total fusion and in situ laser ablation ages. Dating of affected samples by step-heating methods yields mixed $^{40}$Ar/$^{39}$Ar age spectra that might include staircase features at low temperatures and younger ages at medium temperatures. However, appropriate sample preparation and heating schedules can retrieve geologically meaningful ages, including the ages of alteration and original cooling and, as such, dating samples from faults that have experienced polyphase deformation can provide valuable insights into fault deformation.

(2) The main causes of disturbed $^{40}$Ar/$^{39}$Ar age spectra for altered biotite are different total fusion age distributions of sample grains, chemical composition controls on argon release during in vacuo heating, and changes in biotite crystallinity during annealing. Recoil of $^{39}$Ar$_K$ did not have a major effect on the results of this study.

(3) The Waziyü detachment fault underwent prolonged and multiphase deformation associated with hydrothermal fluid activity. K-feldspar $^{40}$Ar/$^{39}$Ar dating revealed rapid cooling at ~120–113 Ma, corresponding to the timing of subsidence of the Fuxin–Yixian Basin. The last stage of activity occurred at ~18–12 Ma and was accompanied by epidotization and muscovitization of biotite. The timing of this event was revealed by $^{40}$Ar/$^{39}$Ar biotite dating using a detailed step-heating method.

**Supplementary Materials:** The following are available online at http://www.mdpi.com/2075-163X/10/8/648/s1, Table S1: $^{40}$Ar/$^{39}$Ar step-heating results, Table S2: $^{40}$Ar/$^{39}$Ar laser results.

**Author Contributions:** Conceptualization, W.S. and W.F.; methodology, W.S.; software, W.S.; validation, W.S., L.Y. and Y.W.; formal analysis, W.S. and Y.W.; investigation, L.W. and F.W.; resources, F.W.; data curation, W.S. and L.Y.; writing—original draft preparation, W.S.; writing—review and editing, W.F.; visualization, W.S.; supervision, W.F. and G.S.; project administration, W.S. and F.W.; funding acquisition, F.W. and W.S. All authors have read and agreed to the published version of the manuscript.

**Funding:** This study is jointly funded by the Ministry of Science and Technology of the People's Republic of China (2016YFC0600109), the National Natural Science Foundation of China (41903050, 41673015), and the Experimental Technology Innovation Fund of the Institute of Geology and Geophysics, Chinese Academy of Science, Grant No.T201904.

**Acknowledgments:** Q. Mao, Y. G. Ma, S. H. Yang, and A. P. Chen were thanked for their assistance during SEM and EMPA work. We thank Gordon Lister for his suggestions during experiment. We thank two reviewers and editors for their constructive comments and suggestions.

**Conflicts of Interest:** The authors declare no conflict of interest.

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
