# Peer review of "Geologically Meaningful 40Ar/39Ar Ages of Altered Biotite from a Polyphase Deformed Shear Zone Obtained by in Vacuo Step-Heating Method: A Case Study of the Waziyü Detachment Fault, Northeast China"

_minerals, doi:10.3390/min10080648_

Round 1
Reviewer 1 Report
The manuscript entitled "Geologically meaningful 40Ar/39Ar ages of altered biotite from a polyphase deformed shear zone obtained by in vacuo step-heating method: A case study of the Waziyü detachment fault, northeast China" discusses the challenges and solutions for dating altered minerals. This study is well-organized and provides important information for geochronologists who particularly work on metamorphic rocks and structural geology.
This manuscript is clear and well-written. The data quality is also good. I suggest a minor revision for this manuscript as "5. Discussion" is a little bit too long and can be more precise.
Overall, this is a high-quality paper.
Author Response
The manuscript entitled "Geologically meaningful 40Ar/39Ar ages of altered biotite from a polyphase deformed shear zone obtained by in vacuo step-heating method: A case study of the Waziyü detachment fault, northeast China" discusses the challenges and solutions for dating altered minerals. This study is well-organized and provides important information for geochronologists who particularly work on metamorphic rocks and structural geology.
This manuscript is clear and well-written. The data quality is also good. I suggest a minor revision for this manuscript as "5. Discussion" is a little bit too long and can be more precise.
Overall, this is a high-quality paper.
Thanks. Yes, the "5. Discussion" part is too long as it also includes the result interpretation. We think it's should keep its integrity under the current state.
Reviewer 2 Report
Comments and suggestions for authors:
Overall I conclude this article to be well written and structured. It is effective in highlighting the problems associated with applying the Ar-Ar technique to products of deformation and the impact of multi-phase deformation on argon systematics. It is important to publish such studies, as it shows that often multiple methods need to be applied to a single sample in order to achieve geologically meaningful age data. That being said, here I make some recommendations before the editor can make the final decision to publish this work.
- It is crucial that all raw data be included in a supplementary data file for every sample analysed, air standards and instrument blanks. It is unacceptable to present geochronological data in a publication without making all raw data files available. Please refer to ‘Data reporting norms for 40Ar/39Ar geochronology (2009) by Renne et al., in Quaternary Geology (4)’.
- At the conclusion of section 3.2 please state what sigma all errors will be reported at (1σ or 2σ) throughout the following text.
- Figure 1: Make reference to inset in the figure caption. Is (A) the small black box in the inset? Please make this clearer and if possible increase figure text size.
- Figure 2: Just for clarity, please add to caption Ep = epidote; Bt = biotite etc.
- Table 2: Please redo this table so that it is a summary of all step heating data for all minerals and expand to include the results of the detailed step heating analyses (line 404 – 405 talks about total fusion ages of CR99NW and CR99M150). Also, CR99-8 K-feldspar is not included. There is no indication what the errors are being reported at, please include this. The caption reads ‘mean ages are calculated from at least two steps’ – please indicate by * or some other notation which mean ages are calculated by just two steps.
- Figure 4: This figure should be expanded to include the step heating age spectra and corresponding argon release pattern of sample CR99M150. This sample has also been subjected to the ‘detailed step heating method’ and I feel that this approach is central to your story and therefor it needs to be included also with CR99NW in figure 4, even though I see it is included in the summary figure 8. The text goes into great detail comparing the two samples, however this would be much easier for the reader to understand if both samples are included in figure 4. Figure 4 states in red ‘Step heating from 750 °C’, however the caption reads ‘The red age spectrum was produced by traditional step-heating up to 750 °C’. Please clarify, I am assuming this is step heating from 750 °C to experiment end (1400 °C – as stated in the text).
- Figure 5, 6, 7 and 8: What are your errors? 1 or 2 σ
Author Response
Overall I conclude this article to be well written and structured. It is effective in highlighting the problems associated with applying the Ar-Ar technique to products of deformation and the impact of multi-phase deformation on argon systematics. It is important to publish such studies, as it shows that often multiple methods need to be applied to a single sample in order to achieve geologically meaningful age data. That being said, here I make some recommendations before the editor can make the final decision to publish this work.
Thanks.
- It is crucial that all raw data be included in a supplementary data file for every sample analysed, air standards and instrument blanks. It is unacceptable to present geochronological data in a publication without making all raw data files available. Please refer to ‘Data reporting norms for 40Ar/39Ar geochronology (2009) by Renne et al., in Quaternary Geology (4)’.
Thanks. All raw data and instrument blanks are included in the supplementary data file. Air standards results are listed as mass discrimination per AMU.
- At the conclusion of section 3.2 please state what sigma all errors will be reported at (1σ or 2σ) throughout the following text.
Thanks. Yes, “Ages uncertainties are reported at 2σ level” is added at the conclusion of section 3.2 to state the age errors.
- Figure 1: Make reference to inset in the figure caption. Is (A) the small black box in the inset? Please make this clearer and if possible increase figure text size.
Thanks. Reference is added to the inset in the figure caption. Text size is increased.
- Figure 2: Just for clarity, please add to caption Ep = epidote; Bt = biotite etc.
Thanks. The meaning of abbreviation in the figure is added.
- Table 2: Please redo this table so that it is a summary of all step heating data for all minerals and expand to include the results of the detailed step heating analyses (line 404 – 405 talks about total fusion ages of CR99NW and CR99M150). Also, CR99-8 K-feldspar is not included. There is no indication what the errors are being reported at, please include this. The caption reads ‘mean ages are calculated from at least two steps’ – please indicate by * or some other notation which mean ages are calculated by just two steps.
Thanks. Result of CR99-8 and detailed step heating analyses of CR99NW and CR99M150 are included in Table 2. Superscript a is used to indicate the mean ages calculated by two steps, and we added this description in the caption.
- Figure 4: This figure should be expanded to include the step heating age spectra and corresponding argon release pattern of sample CR99M150. This sample has also been subjected to the ‘detailed step heating method’ and I feel that this approach is central to your story and therefor it needs to be included also with CR99NW in figure 4, even though I see it is included in the summary figure 8. The text goes into great detail comparing the two samples, however this would be much easier for the reader to understand if both samples are included in figure 4. Figure 4 states in red ‘Step heating from 750 °C’, however the caption reads ‘The red age spectrum was produced by traditional step-heating up to 750 °C’. Please clarify, I am assuming this is step heating from 750 °C to experiment end (1400 °C – as stated in the text).
Thanks. Yes, the step-heating result of CR99M150 is included in figure 4. The description of “ step-heating up to 750°C “ in the caption was a mistake, and we corrected it in this version.
- Figure 5, 6, 7 and 8: What are your errors? 1 or 2 σ
Thanks. Age errors are plotted as 2 σ, and state is added in the text.